# Genetically Engineered Artificial Microvesicles Carrying Nerve Growth Factor Restrains the Progression of Autoimmune Encephalomyelitis in an Experimental Mouse Model

**DOI:** 10.3390/ijms24098332

**Published:** 2023-05-05

**Authors:** Reem Alatrash, Maria Golubenko, Ekaterina Martynova, Ekaterina Garanina, Yana Mukhamedshina, Svetlana Khaiboullina, Albert Rizvanov, Ilnur Salafutdinov, Svetlana Arkhipova

**Affiliations:** 1Institute of Fundamental Medicine and Biology, Kazan (Volga Region) Federal University, 420008 Kazan, Russia; alatrash.reem95@gmail.com (R.A.); golubenckomasha@yandex.ru (M.G.); ignietferro.venivedivici@gmail.com (E.M.); rizvanov@gmail.com (A.R.); svetlanaarkhipva@yandex.ru (S.A.); 2Department of Medical Biology and Genetics, Kazan State Medical University, 420012 Kazan, Russia

**Keywords:** neurodegeneration, experimental autoimmune encephalomyelitis (EAE), multiple sclerosis, lentivirus, NGF, MOG, ADMSC, artificial microvesicles, cytochalasin B, astrocytes, demyelination, cytokines

## Abstract

Multiple sclerosis (MS) is an incurable, progressive chronic autoimmune demyelinating disease. Therapy for MS is based on slowing down the processes of neurodegeneration and suppressing the immune system of patients. MS is accompanied by inflammation, axon-degeneration and neurogliosis in the central nervous system. One of the directions for a new effective treatment for MS is cellular, subcellular, as well as gene therapy. We investigated the therapeutic potential of adipose mesenchymal stem cell (ADMSC) derived, cytochalasin B induced artificial microvesicles (MVs) expressing nerve growth factor (NGF) on a mouse model of multiple sclerosis experimental autoimmune encephalomyelitis (EAE). These ADMSC-MVs-NGF were tested using histological, immunocytochemical and molecular genetic methods after being injected into the tail vein of animals on the 14th and 21st days post EAE induction. ADMSC-MVs-NGF contained the target protein inside the cytoplasm. Their injection into the caudal vein led to a significant decrease in neurogliosis at the 14th and 21st days post EAE induction. Artificial ADMSC-MVs-NGF stimulate axon regeneration and can modulate gliosis in the EAE model.

## 1. Introduction

Multiple sclerosis (MS) is an autoimmune disease characterized by lesions in the central nervous system (CNS) and substantial axon loss [1]. These lesions are believed to result from demyelination accompanied by chronic inflammation [2,3]. Inflammation is evident via perivascular infiltration of T and B lymphocytes serving as a source of bioactive mediators diffusing into the brain parenchyma [4]. These mediators include pro-inflammatory cytokines, chemokines and adhesion molecules [5,6], which can recruit more leukocytes and activate neuroglia [7]. Reactive astrocytes (RAs), a subset of neuroglia in CNS, could contribute to myelin degeneration and facilitate leukocytes’ migration to the brain [8,9].

Inflammation characterized by T lymphocytes and macrophage infiltration was suggested as playing the central role in the oligodendrocyte’s apoptosis and degeneration of the myelin sheath [3]. This obliteration of myelin initiates axon disintegration and formation of plaques, areas of the brain lesions in the CNS [10]. These lesions are characterized by the death of the brain cells, which manifests with clinical signs of the brain damage such as impaired cognitive function and physical disability in MS [11,12]. MS disabilities are also associated with severe functional limitations affecting daily and professional activities and socialization [13,14].

Currently, the therapy for MS is based on preventing neurodegeneration and suppressing the aberrant immune response [15]. Immunomodulatory drugs used to treat MS reduce disease symptoms and prolong remission [16]. However, existing strategies have limited efficacy, failing to avert symptoms worsening. Cell therapy and secretome modulation have become prospective approaches in MS treatment [17,18]. Several studies have demonstrated the therapeutic efficacy of mesenchymal stem cells (MSCs) in MS due to their immunomodulatory and regenerative properties [19,20,21,22]. It was suggested that cytokines and growth factors secreted by MSCs stimulate tissue regeneration [23] via a paracrine mechanism [24,25,26]. In addition, a cell-free extracellular-vesicle (EV)-based therapy was suggested as an alternative to the use of MSCs [27,28,29,30]. EVs produced in the form of the exosomes, apoptotic bodies and microvesicles (MVs) are secreted under physiological and pathological conditions [31,32,33] to be used for cell communication [34,35].

MSCs could be a source of EVs with cargo-containing molecules expressed in these cells. MSCs constitutively expressing and releasing nerve growth factor (NGF), a neuroprotective molecule controlling neurons’ survival and differentiation [36,37,38], could be used as therapeutics for the treatment of MS. NGF’s therapeutic effect in MS could include promoting myelin synthesis with oligodendrocytes and Schwann cells [39,40,41]. This could restore the axon myelination affected by MS [2,42,43]. Together with promoting myelin production, NGF has immunomodulatory properties that could reduce inflammation and activate the aberrant immune response [44,45,46,47].

We have addressed the pressing need for novel approaches to MS treatment. To address this urgent need, we sought to determine the therapeutic efficacy of the adipose-derived MSCs (ADMSCs) and ADMSC-derived MVs (ADMSC-MVs) expressing NGF in mice with an experimental autoimmune encephalomyelitis (EAE), a model of MS [48,49]. We have found that the local delivery of NGF packed inside MVs to the EAE spinal cord has a therapeutic potency by enhancing the remyelination of axons, improving the EAE spinal cord tissue structure and reducing the size of the demyelinated areas. Additionally, our data showed that MVs expressing NGF could have an immunomodulating effect on serum cytokine levels.

## 2. Results

### 2.1. Characterization of ADMSC-MVs and Analysis of NGF Expression

Morphological analysis of ADMSCs showed spindle-shaped or polygonal cells (Figure 1A). The nucleus was elongated with the presence of euchromatin and a thin border of heterochromatin along the periphery. In the cytoplasm of ADMSCs, in addition to various organelles, there were many membrane-limited microvesicles (*). Additionally, several membrane-limited natural exosomes were revealed in the immediate vicinity of the cells (arrow). After transduction with NGF, the cell ultrastructure did not undergo strong changes. (Figure 1A′) The success of transduction was judged by the presence of GFP green glow after immunocytochemical staining of cells with anti-NGF antibodies (Figure 1B,C).

Analysis of ADMSC-MVs revealed a set of the double-membrane structure of spherical or irregular shape with a diameter of 50–200 nm (Figure 1D). The early size, membrane structure, immune phenotype and proteome profile were characteristic of MVs [50,51,52].

Next, we demonstrated NGF protein in ADMSC-MVs-NGF cargo via WB (Figure 2E). Total proteins (10 µg) from ADMSC-MVs-NGF were employed for the analysis. MVs from non-transduced ADMSCs were used as control. NGF protein was visualized only in ADMSC-MVs-NGF, while the ADMSC-MVs did not contain NGF. These data confirm the expression of recombinant NGF in ADMSC-MVs-NGF.

Additionally, the location of NGF in generated ADMSC-MVs-NGF was demonstrated by ICC using secondary antibodies conjugated with gold nanoparticles (5 nm diameter). NGF was found in the cytoplasm of ADMSC-MVs-NGF (arrows), as well as in the vicinity and on the membranes (Figure 1F,G). NGF was rarely captured outside the cytoplasm, probably due to damage to individual microvesicles.

### 2.2. Morphological and Quantitative Analysis of Spinal Cord Neurogliosis

Morphological analysis of glial cells in experimental groups was performed (Figure 2). Electron microscopy of glial cells in EAE groups revealed cells with ultrastructures characterized by a marked increase in the number of glycogen granules and mitochondria and high electron density of the cytoplasm (Figure 2A). These cells were primarily astrocytes that showed hypertrophy and an increased volume of cytoplasm filled with fibrils.

Semi-thin cross-sections of the spinal cord tissues stained with methylene blue were viewed and analyzed under a light microscope. In tissues obtained from a group of untreated animals, an unusual proliferation of individual glial cells was observed with areas of destruction and demyelination of the tissue (Figure 2B,C). Those cells had hypertrophic morphology characterized by a massive increase in cell soma with dark staining and reduced density of processes.

The number of glial cells was significantly reduced in all treated mice as compared with that in untreated mice (*p* < 0.05) (Figure 2D–G′,J). After treatment with ADMSC-MVs, the numeral of glial cells was significantly reduced as compared with that without treatment (*p* < 0.05). At 21 dpi, the average number of cells was significantly lower than that at 14 dpi. After treatment with ADMSC-MVs-NGF, the number of glial cells was significantly reduced as compared with that without treatment (*p* < 0.05). However, there was no significant difference in glial cell count between 14 and 21 dpi (Figure 2G).

### 2.3. TEM Analysis

TEM analysis of tissue samples was conducted from the following experimental groups: 1. Control without EAE; 2. EAE without treatment; 3. EAE treated with ADMSCs; 4. EAE treated with ADMSC-MVs; 5. EAE treated with MVs-NGF.

#### 2.3.1. Control Animals

In the anterior horns gray matter of the spinal cord of controls (Figure 3A), the motor neurons had a round nucleus, nucleolus and a cytoplasm with well-developed organelles. Neuronal perikaryons were localized between the myelinated (arrow) and myelin-free processes of other neurons. Nerve processes formed clearly visible synaptic contacts between themselves (Figure 3D). Astrocytes in this area appeared to have a cytoplasm of medium electron density, evident organelles and an irregularly shaped nucleus with heterochromatin. Astrocytes did not exceed 10 µm in diameter, indicating that they were not reactive (Figure 3B). Additionally, oligodendrocytes were identified between nerve fibers with an oval nucleus characterized by peripheral localization of heterochromatin, an electron-transparent cytoplasm and evident organelles (Figure 3C).

#### 2.3.2. EAE

In contrast, the gray matter of the spinal cord in mice with EAE had wide cavities (Figure 4A (*)). Additionally, the cavity edges had disintegrated unidentifiable substances, demyelinated axons and cells with lobed nuclei (Figure 4A, green arrow). These disintegrated areas were surrounded by preserved sites, along with clusters of RAs. Moreover, there were small areas with neurons having degenerated (Figure 4A (*)) myelin and unmyelinated axons (Figure 4B,E). The lack of synapses between cells was characteristic of these areas. RAs showed electron-dense cytoplasms with more than 20 µm in length and irregular or pyramidal shapes. They also had a cytoplasm rich with free ribosomes, defined Golgi apparatus and numerous tanks of rough ER (Figure 4E′). These cells had numerous round-shaped mitochondria (Figure 4E″) with well-developed cristae. Microglial cells were also detected in the partially disintegrated area (Figure 4C), as well as degenerated oligodendrocytes with cytoplasms lacking organelles (Figure 4D). It is believed that reactive glia and their processes form a glial scar between the preserved zones and the zones of disintegration, probably similar to the astroglia barrier in the injured spinal cord [53].

#### 2.3.3. EAE Treated with ADMSCs

Ultrastructural analysis of the gray matter of the spinal cord of an EAE mouse treated with ADMSCs at 14 dpi showed that gray matter tissues remain predominantly intact (Figure 5A). Only small cleared cavities were visualized in place of individual degenerated fibers (*). Large cavities resulting from demyelination and degeneration were not visualized. Conversely, there were single fibers in a state of remyelination (arrow) (Figure 5B). However, isolated degenerating motor neurons (highlighted in pink) at 21 dpi were identified (Figure 5B). Individual degenerated fibers appeared as clear areas surrounded by a reactive astrocyte group that formed the border between the intact area and the area with individual degenerated fibers at 14 dpi (Figure 5C,C′, selected area). The photo shows the ultrastructure of a reactive astrocyte (highlighted in green) and adjacent tissue with signs of degeneration: clearing and lack of organelles. A reactive astrocyte has a more electron-dense cytoplasm and richly developed organelles, such as the Golgi apparatus, phagolysosomes, mitochondria and ER, which indicates the phagocytic activity of these cells, as well as the biosynthesis of proteins, probably cytokines, and growth factors.

#### 2.3.4. EAE Treated with ADMSC-MVs

A fine structure study of the gray matter of the spinal cord of EAE mice treated with ADMSC-MVs showed the presence of preserved myelin fibers (arrow) and oligodendrocytes (OL). (Figure 6A). However, small, enlightened areas of degeneration, in the area in which neutrophils (Neu) were present, were detected at 14 dpi (Figure 6A). Additionally, in this experimental group, neurons (N) were detected in the initial stage of degeneration with vacuolized cytoplasms. The perikaryon was surrounded by cells with an electron-dense cytoplasm, probably reactive astrocytes (Figure 6B: RAs, highlighted in green.) In addition, a preserved neuron in a modified tissue with an adjacent preserved oligodendrocyte was visualized (Figure 6C). At 14 dpi, the modified area showed a homogeneous preserved tissue along with small regions of degenerated nerve processes. Preserved tissues also revealed preserved neurons similar in structure to neurons in modified areas (highlighted in pink) at 21 dpi (Figure 6D). A reactive astrocyte at 21 dpi in an area with many degenerated lucid axons was also seen (Figure 6E).

#### 2.3.5. EAE-MVs-NGF

Ultrastructure analysis of the gray matter of the spinal cord in EAE mice receiving ADMSC-MVs-NGF demonstrated the absence of cavities. Additionally, by day 14 post EAE induction, mice receiving ADMSC-MVs-NGF showed signs of remyelination (Figure 7A; arrow). Additionally, on day 21, islands of newly remyelinated fibers were found (shown in Figure 7B). These remyelinated islands appeared in the areas of gray matter where they are usually not present (Figure 7B).

Along with myelin fibers, there were myelin-free fibers with many organelles, in particular mitochondria, but synapses between fibers were absent. Small, degenerated axons were also present (Figure 7A (*)). Interestingly, this remyelination and restoration of neural tissue was accompanied by the presence of an RA (Figure 7D; green). This RA had an altered morphology and a cytoplasm rich with organelles and round mitochondria, with an increased overall size of the astrocyte cell (shown in Figure 7C), similar to those in EAE. Additionally, lymphocytes (Figure 7D,E; yellow) were detected in those areas. Motor neurons remained morphologically identical to those in non-EAE controls at 14 and 21 days after EAE induction, regardless of treatment status (shown in Figure 7D,E).

### 2.4. Measurement of Demyelination Area

A morphological analysis of tissue specimens collected at 14 dpi and 21 dpi after the induction of EAE was also completed by measuring the size of the demyelinated area.

Semi-thin cross-sections (1 μm) of the spinal cord tissue were viewed under a light microscope after staining with methylene blue. EAE spinal cord tissue without therapy had several areas of demyelination in the white matter on day 14.

Additionally, single areas of destruction in the gray matter appeared as a thin longitudinal cavity with clusters of RAs (Figure 8A). On day 21, areas of destruction in both white and gray matter increased in size (Figure 8A,F).

ADMSC treatment prevented demyelination of white matter on day 14 after EAE induction. (Figure 8B,F). However, in the gray matter of the spinal cord, destruction with signs of cell death was still found on day 14. On day 21, wide areas of degenerated tissues were found without cavities in the EAE-received ADMSCs (Figure 8B,F).

A significant decrease in the size of the demyelination area was found between the EAE mice receiving ADMSC-MVs and EAE without treatment on both days 14 and 21 (*p* < 0.05) (Figure 8C,F). However, there was no significance in reducing the demyelinated area after MVs-NGF administration compared to ADMSC-treated EAE.

EAE mice receiving ADMSC-MVs-NGF had a significantly smaller demyelination area than untreated EAE and ADMSC-MVs treated mice on both days 14 and 21 after EAE induction (Figure 8D,F).

### 2.5. Cytokine Profile Evaluation in Blood Serum

Cytokine levels were analyzed in the serum samples from experimental groups on the 21st day after the formation of EAE: 1. Control without EAE mice, 2. EAE mice, 3. EAE treated with ADMSCs, 4. EAE treated with ADMSC-MVs, 5. EAE treated with ADMSC-MVs-NGF (Figure 9A).

Serum level of TNF-α was increased in untreated EAE mice compared with controls without EAE. However, this cytokine level was reduced after treatment with ADMSC-MVs and ADMSC-MVs-NGF (Figure 9B). Similar changes were found in the serum level of IL-6, which increased in EAE compared with controls without EAE. Administration of ADMSC-MVs and ADMSC-MVs-NGF reduced the level of this cytokine in EAE (shown in Figure 9B). Additionally, CCL2, CCL5 and IL-1β increased in EAE and were significantly reduced after treatment with ADMSC-MVs and ADMSC-MVs-NGF. Interestingly increased levels of IL-10 were found in EAE mice treated with ADMSC-MVs and ADMSC-MVs-NGF. IL-10 in the treated groups was even higher than in EAE without treatment (Figure 9B). Levels of IL-4 levels were moderately increased in mice treated with ADMSC-MVs-NGF, which was still high significantly higher than that in EAE without treatment (shown in Figure 9B).

## 3. Discussion

We have demonstrated profound changes in mice’s brains and spinal cord structures with EAE. An increased RA count was found in the brain and spinal cord of EAE. Detection of RAs was evident by the upregulation of GFAP, increased cell size and large number of organelles. An increased expression of GFAP was the primary marker of astrogliosis [54,55]. This astrogliosis is a unique feature of CNS pathology in MS [56], viewed as a neuroglial response [57,58]. Astrocyte activation is commonly found in EAE, which presents with upregulation of GFAP, hypertrophy, proliferation, changes in gene and protein expression and scar formation [59,60]. This is also accompanied by the loss of terminal processes around small blood vessels and increased blood–brain barrier (BBB) permeability, leading to CNS inflammation and perivascular edema [61,62]. RAs could also produce pro-inflammatory cytokines and reactive oxygen species [63].

These changes in the CNS structure were reduced in EAE after treatment with ADMSC-MVs-NGF. We have shown that this treatment decreased demyelination in the spinal cord and alleviated tissue damage. Decreased demyelination is a sign of successful treatment [64,65]. NGF regulates remyelination by promoting oligodendrocytes’ and Shawn cells’ proliferation and activity [66]. Our results demonstrate that the potency of NGF remained preserved when it was administrated in the cargo of ADMSC-MVs-NGF. The effect of NGF was evident by demonstrating the signs of remyelination, as a formation of the new myelin sheaths was shown by TEM. Additionally, the neuron structure was preserved, as shown in the ultrastructure examination of tissues after administering ADMSC-MVs-NGF. These results correlate with Wang et al. They demonstrated the therapeutic effect of the modified neural stem cells overexpressing NGF (NGF-NSCs) in the spinal cord injury model [37]. In this study, the electron microscopy analysis of the white matter in the lesion epicenter showed that the integrity of the nerve sheath and corresponding axons were preserved after NGF-NSCs transplantation. NGF-NSCs reduced the oligodendrocyte loss, attenuated astrocytosis, reduced demyelination and kept the motor neurons [37].

We suggest that NGF contributes to the process of remyelination and tissue repair. This observation is consistent with the work of Villoslada et al. [47]. The authors demonstrated decreased expression of histological indicators of inflammation and significantly reduced demyelination. Additionally, decreased IFN-γ with increased IL-10 production by microglial cells in inflammatory EAE lesions was found after treatment with ADMSCs-NGF. Villoslada et al. have also provided evidence that continuous intracerebral infusions of recombinant human NGF reduce the severity of EAE in non-human primates [47]. However, the level of NGF required for therapeutic effect remains unknown [67]. It could be concluded that the application of MVs as the delivery vehicle for NGF contributes to therapeutic outcomes leading to remyelination at the early stage (day 14).

It should be noted that the therapeutic efficacy of ADMSC-MVs was less pronounced as compared with the effect of ADMSC-MVs-NGF. However, the potency of ADMSC-MVs alone was demonstrated. We found a significant improvement in histological features and the presence of the new myelin sheaths compared with EAE without treatment. Our data support the previous finding of Jafarinia et al., where the therapeutic effects of natural extracellular MSC vesicles in EAE was described [68]. Authors have shown decreased inflammation and demyelination in EAE mice treated with MSC vesicles. We suggest that the therapeutic effect of ADMSC-MVs and ADMSC-MVs-NGF is linked to reduced demyelination.

Inflammatory cytokine TNF-α was shown to contribute to the pathogenesis of EAE [69]. Ruddle et al. have demonstrated that neutralizing this cytokine reduces symptoms of EAE. Additionally, chemokines such as CCL2, CCL3, CCL4, CCL5 and CCL11 were shown to recruit autoreactive glial cells and immune cells from the periphery to the CNS secretion of pro-inflammatory cytokines, contributing to demyelination and loss of neurons [70]. Therefore, analysis of these cytokine and chemokine serum levels could help determine the mechanisms of ADMSC-MVs’ and ADMSC-MVs-NGF’s therapeutic efficacy.

We have found that ADMSC-MVs-NGF moderately increased IL-10 levels. IL-10, produced by T-helpers type 2 [71,72], is a potent anti-inflammatory cytokine that plays a critical role in preventing inflammation and autoimmune response [73,74]. It was shown that astrocytes are the primary source of IL-10 in MS lesions [75,76]. According to Ransohoff et al., secretion of this cytokine by astrocytes is mediated by NGF [67]. It was suggested that the high level of IL-10 expression by parenchymal astrocytes was directly regulated by NGF [77]. This may explain our findings of unique IL-10 activation in the ADMSC-MVs-NGF treated EAE. IL-10 is believed to attenuate astroglial reactivity [78] and induce microglial changes in a beneficial effect on neuronal survival [74]. It also acts via peripheral immunosuppressive effect to attenuate EAE [79].

We used ADMSCs as a therapeutic control for ADMSC-MVs-NGF. Our results showed that ADMSCs could minimize destruction and demyelination compared with that of ADMSC-MVs. This finding is consistent with the data of Conforti et al., which stated that MVs have a lower in vitro immunomodulatory effect on T-cell proliferation and antibody formation than their cellular counterparts [80]. This can be explained by the fact that MSCs, when administrated in vivo, can proliferate and secrete anti-inflammatory cytokines and growth factors as compared with MVs [81]. However, MVs are a better substitution for cell-based therapeutics [82]. Additionally, it should be considered that the MSCs’ limitations are related to their potential tumorigenicity [83,84] and resistance to apoptosis [85].

The neuroprotective activity of NGF, together with its immunomodulatory effect [86], make this protein an attractive candidate for the treatment of MS [87]. Delivery of this protein in a cargo of ADMSC-MVs could be a reliable and effective approach for treating this disease. That is especially important, as recombinant NGF was shown to fail to crossing the BBB [88,89,90]. Therefore, ADMSC-MVs capable of crossing BBB could deliver NGF into the brain tissue. In addition, when administered systemically, NGF can cause serious side effects due to its pleiotropic properties [86,91,92]. This side effect could also be overcome, as ADMSC-MVs retain NGF until its delivery to the target organ. The difficulties encountered in the clinical application of NGF urge us to explore novel mechanisms of delivery to the brain, where ADMSC-MVs could provide the solution.

## 4. Materials and Methods

### 4.1. Isolation and Maintenance of ADMSCs

ADMSCs were isolated from the murine adipose tissue via enzymatic digestion with 0.2% collagenase II (BioLot, Moscow, Russia) for 1 h at 37 °C under constant shaking at 180 rpm [93]. ADMSCs were maintained using complete α-MEM medium supplemented with 10% FBS, 4mM L-glutamine and 1% streptomycin and penicillin. All supplements were purchased from PanEco (Moscow, Russia).

### 4.2. Recombinant Lentivirus Expressing Nerve Growth Factor (LV-NGF)

HEK293T cells were seeded on a 10 cm culture plate (2–2.5 × 10^6^ in 10 mL) and maintained (37 °C, 5% CO_2_) using complete DMEM medium supplemented with 10% FBS, 4 mM L-glutamine and 1% of streptomycin and penicillin (PanEco, Moscow, Russia) until they reached 70% density. Medium was replaced and cells were transfected with 500 µL of a DNA mix containing: 14 μg vector plasmid pLX-NGF-10 or pLX-Katushka2S; 6.63 μg packaging plasmid psPAX2; 3.51 μg envelope plasmid pCMV-VSVG; 50 μL 2.5 M CaCl_2_; 293 μL 0.1 × TE (10 mm Tris + 1 mm EDTA pH 8.0); and 151 μL of H_2_O. Then, 506 µL of 2× Hepes Buffer Saline (HBS) (0.280 M NaCl, 0.1 M Hepes, 0.0015 M Na_2_HPO_4_, pH 7.12) was added to the mix, vortexed and incubated at room temperature for 25 min. The compound was added dropwise onto the cell monolayer and incubated for 6–8 h at 37 °C, 5% CO_2_. Efficacy of transfection was assessed using fluorescent microscope AxyObzerver Z.1 (Carl Zeiss, Jena, Germany). scanning red fluorescent channel. Cells transfected with pLX-Katushka2S expressing red fluorescent protein were used as control for transfection.

Supernatants containing recombinant lentiviruses LV-NGF or LV-Katushka-2S were collected 24 h after transfection and fresh medium was added. The virus-containing supernatants were harvested every 12 h for 3 days. Combined, supernatants were cleared (5 min at 1500 rpm), filtered (0.22 μm filter) and concentrated (26,000 rpm for 2 h at 4 °C) using an Optima ™ L-90K (Beckman Coulter, Brea, CA, USA) ultracentrifuge. Supernatant was removed and the lentivirus containing precipitate was resuspended in a serum-free medium. Virus aliquots of 2 mL were stored at −80 °C.

### 4.3. ADMSCs Expressing NGF

ADMSCs were plated in a 6-well plate and maintained until they reached 50% confluence. Cells were transduced with LV-NGF with multiplicity of infection (MOI100). Cells transduced with lentivirus pKatushka2S (MOI100) were used as control. Transduced cells were incubated for 3 days (37 °C, 5%, CO_2_) before harvesting MVs.

### 4.4. Harvesting of ADMSC-MVs

ADMSC-MVs were produced according to Pick et al. with some modifications [94]. ADMSCs were incubated in serum-free DMEM containing 10 µg/mL of cytochalasin B (Sigma-Aldrich, St. Louis, MO, USA) for 30 min at 37 °C and 5% CO_2_ before active stirring for 30 s. MVs were isolated following series of centrifugations: 500 rpm for 10 min, 700 rpm for 10 min and 4000 rpm for 25 min. The pellet containing ADMSC-MVs was washed (PBS, Panico, Moscow, Russia) and pelleted at 4000 rpm for 25 min. ADMSC-MVs expressing NGF were designated as ADMSC-MVs-NGF.

### 4.5. Immunocytochemistry (ICC) 

ADMSC-MVs and ADMSC-MVs-NGF were pelleted on 12,000 RPM in DPBS solution (PanEco, Moscow, Russia), fixed (10% buffered formaldehyde; Alfa Aesar, Thermo Fisher Scientific, Dreieich, Germany) at 4 °C for 12 h, followed by post-fixation using 0.5% OsO4 (Sigma-Aldrich, St. Louis, MO, USA) for 30 min at RT. Dehydrated using ethanol gradient, samples were embedded in LR White polymer resin (Electron Microscopy Sciences, Hatfield, PA, USA). Ultrathin sections (0.1 μm) were made using a Leica UC7 ultramicrotome (Leica, Wetzlar, Germany). Ultra-thin sections were mounted onto formvar-coated nickel grids and blocked with 10% normal donkey serum, bovine serum albumin 0.2% and 0.1% Triton X-100 in Tris buffer solution (Tris 0.01 M, NaCl 0.15 M pH = 8.2) for 1 h at RT. Washing with TBS, sections were incubated overnight at 4 °C with anti-NGF antibodies (1:100; Sigma-Aldrich, N8912) followed by the secondary antibodies conjugated with colloidal gold (diameter of gold nanoparticles 5 nm) (Sigma-Aldrich, St. Louis, MO, USA) for 2 h at RT. Then, sections were incubated in a Silver enhancing kit (BBI Solutions, Caerphilly, UK) for 2 min at RT in the dark followed by washing with TBS (pH 8.2). Sections were contrasted with uranyl acetate and lead citrate and analyzed using a transmission electron microscope (Hitachi HT7700, Hitachi, Ltd., Chiyoda City, Japan).

### 4.6. Western Blotting (WB)

ADMSC-MVs or ADMSC-MVs-NGF were collected using lysis buffer (50 mM HEPES, 150 mM NaCl, 1 mM EDTA; 0.5% Triton-X100, 10% glycerol, Panreac, Barcelona, Spain) containing Halts protease inhibitor (1 μM; Therscientific, Rockford, IL, USA) for 30 min and pelleted (12,000× *g*). Total protein concentration was determined using BCA Protein Assay Kit (Thermo Fisher Scientific). Proteins (10 μg/well) were separated using 4–12% gradient SDS-PAGE gel and transferred onto a polyvinylidene fluoride (PVDF) membrane (Bio-Rad Laboratories, Hercules, CA, USA). Membrane was blocked (5% non-fat dry milk in PBS with Tween-20 (PBS-T), pH 7.4)) for 1 h at room temperature and probed with primary rabbit anti-NGF antibodies (1:100; Santa Cruz, Dallas, TX, USA, sc-548) at 4 °C for 18 h followed by the secondary anti-rabbit HRP-conjugated secondary antibodies (1:1000; Thermo Fisher Scientific, Dreieich, Germany) for 2 h at RT (shown in Table 1). Antibody reaction was visualized using the Clarity Western ECL Substrate kit (Bio-Rad Laboratories). Analysis and normalization of protein load was conducted using Image Lab software version 4.1 (Bio-Rad Laboratories).

### 4.7. Immunofluorescence Histochemistry

Tissue cross-sections were blocked with 5% goat serum for 1 h at room temperature and incubated overnight at 4 °C with primary Abs (Table 1). Washed (3×, PBS), sections were incubated with fluorophore-conjugated secondary Abs for 2 h at room temperature. 4′,6-Diamidino-2-phenylindole (DAPI) (10 μg/mL in PBS, Sigma) was used to stain the nucleus. Stained tissue sections were analyzed using an LSM 780 Confocal Microscope (Carl Zeiss, Oberkochen, Germany). Images were captured in the z-plane using identical confocal settings (laser intensity, gain and offset). Measurements were obtained from transverse histological sections collected at Th8 vertebral level.

### 4.8. Transmission Electron Microscopy (TEM)

Tissue samples were fixed (2.5% phosphate-buffered glutaraldehyde solution; Merck, Kenilworth, NJ, USA) at 4 °C for 24 h, postfixed (1% OsO4; Electron Microscopy Sciences, Hatfield, PA, USA) for 1 h at RT, dehydrated using an ethanol gradient and embedded in embedding medium EPON 812 (Electron Microscopy Sciences, Hatfield, PA, USA). Ultrathin (0.1 µm) sections were stained with uranyl acetate and lead citrate before analysis using a transmission electron microscope (Hitachi HT7700, Tokyo, Japan).

### 4.9. Animal Housing

Female C57BL/6 mice (6–7 weeks of age, 15–17 g weight) (Pushchino, Moscow, Russia) were used in experiments. Mice were housed at the Kazan Federal University animal facility. All animals received food and water ad libitum and were contained in a room with a 12 h light/12 h dark cycle and an ambient temperature of 22 °C. All experiments were carried out to comply with the procedure protocols approved by Kazan Federal University local ethics committee (protocol No, date 27 May 2014) according to the rules adopted by Kazan Federal University and Russian Federation Laws.

### 4.10. Inducing of EAE Mouse Models

To induce EAE, 100 mg of dry Mycobacterium tuberculosis was mixed with 10 mL of an incomplete Freund adjuvant (IFA) to prepare complete Freund adjuvant (CFA). An equal volume of myelin oligodendrocyte glycoprotein 35–55 (MOG) (2 mg/mL in PBS) and CFA were mixed to make an emulsion. EAE was induced via injection of 100 µL of MOG + CFA emulsion subcutaneously into different sites on each hind flank near the lymph nodes on both sides. Immediately after MOG + CFA injection, Bordetella pertussis toxin (PTX) (5 ng/mL, 100 µL; Tocris Bioscience, Bristol, UK) was injected into the tail vein. A second intraperitoneal injection of PTX (100 µL) was performed 48 h later.

Animals were randomized into 5 groups with 8 mice per group: (1) negative control (EAE without treatment), (2) EAE treated with ADMSCs, (3) EAE treated with ADMSC-MVs, (4) EAE treated with ADMSC-MVs-NGF, (5) control animals without EAE or ADMSC-MVs treatment.

Animals were assessed for clinical signs of EAE each day after induction until sacrifice. Severity of the symptoms was evaluated using a scale ranging from 0 to 5: grade 0 = no signs, grade 1 = partial loss of tail tonicity, grade 2 = total loss of tail tonicity, grade 3 = unsteady gait and mild paralysis, grade 4 = hind limb paralysis and incontinence and grade 5 = moribund or death. EAE was diagnosed by mice having a score that exceeded 2. Mice that had a score lower than 2 were not included in the experimental design. EAE severity was assessed before the day of sacrifice where all animals had grades between 2 and 3. In the non-EAE control group, scores remained as grade 0. The onset of clinical symptoms of EAE was on day 7. MVs and MVs-NGF were administrated intravenously through tail injection. Organs and blood samples were collected on days 7 and 14 after the initiation of the treatment (day 14 and 21 post immunization (dpi), respectively). After anesthesia, mice were sacrificed with chloral hydrate (125 µL of 6.4% solution). Treatment protocol and sample collection are summarized in Figure 10.

### 4.11. Cytokine Analysis

Serum cytokines were analyzed using Bio-Plex Pro Mouse Cytokine 23-plex Assay #M60-009RDPD (BioRad), enabling simultaneous detection of 23 cytokines/chemokines/interleukins (IL-1alpha, IL-1beta, IL-2, IL-3, IL-4, IL-5, IL-6, IL-9, IL-10, IL-12 (P40), Il-12 (P70), Il-13, Il-17A, eotaxin (CCL11), G-CSF, GM-CSF, IFN-gamma (IFNγ), KC, MCP-1 (CCL2), MIP-1alpha (CCL3), MIP-1beta (CCL4), RANTES (CCL5), TNF-alpha (TNFα). Serum aliquots (50 μL) were analyzed with a minimum of 50 beads acquired per analyte. Median fluorescence intensities were collected using a Luminex 100 or 200 analyzer (Luminex, Austin, TX, USA). Each sample was analyzed in triplicate and the resulting data were analyzed with MasterPlex CT control software v.3 and MasterPlex QT analysis software v.3 (MiraiBio, San Bruno, CA, USA). Standard curves for each cytokine were generated using standards provided by the manufacturer. Data were analyzed using MasterPlex CT control software v.3 and Master-Plex QT analysis software v.3 (MiraiBio, Alameda, CA, USA). A standard curve was created for each analyte. The results of multiple fluorescence intensity were gathered and calculated using BioPlex Manager V.6.1 (BioRad).

### 4.12. Morphometric Analysis 

Morphometric analysis of the demyelination area in the spinal cord sections (1 μm) was conducted using the Aperio ImageScope 12.0.1 software; Zen 2012 software (Carl Zeiss) was used to measure the diameter of ADMSC-MVs.

### 4.13. Statistical Analysis

Statistical analysis was performed using the non-parametric Kruskal–Wallis test using IBM SPSS statistics 26.0 (IBM Corporation, Armonk, NY, USA). The experiments were performed in three technical repetitions. For all statistical data, the confidence level was accepted as less than 0.05 (p < 0.05).

## 5. Conclusions

In the current study, we demonstrated the therapeutic potential of ADMSC-MVs-NGF in an EAE mice model of MS. ADMSC-MVs-NGF showed therapeutic efficacy demonstrated using multiple criteria such as smaller area of spinal cord tissue degeneration and a lower level of serum pro-inflammatory cytokines. Thus, the approach using gene-cell vesicular therapy with ADMSC-MVs-NGF expression proved to be a novel approach for the treatment of MS. Future studies should also focus on expanding the current understanding of ADMSC-MVs-NGF use in therapy for MS, storage and reconstitution of artificial vesicles and incorporation of other MS models to evaluating behavioral responses.

## Figures and Tables

**Figure 1 ijms-24-08332-f001:**
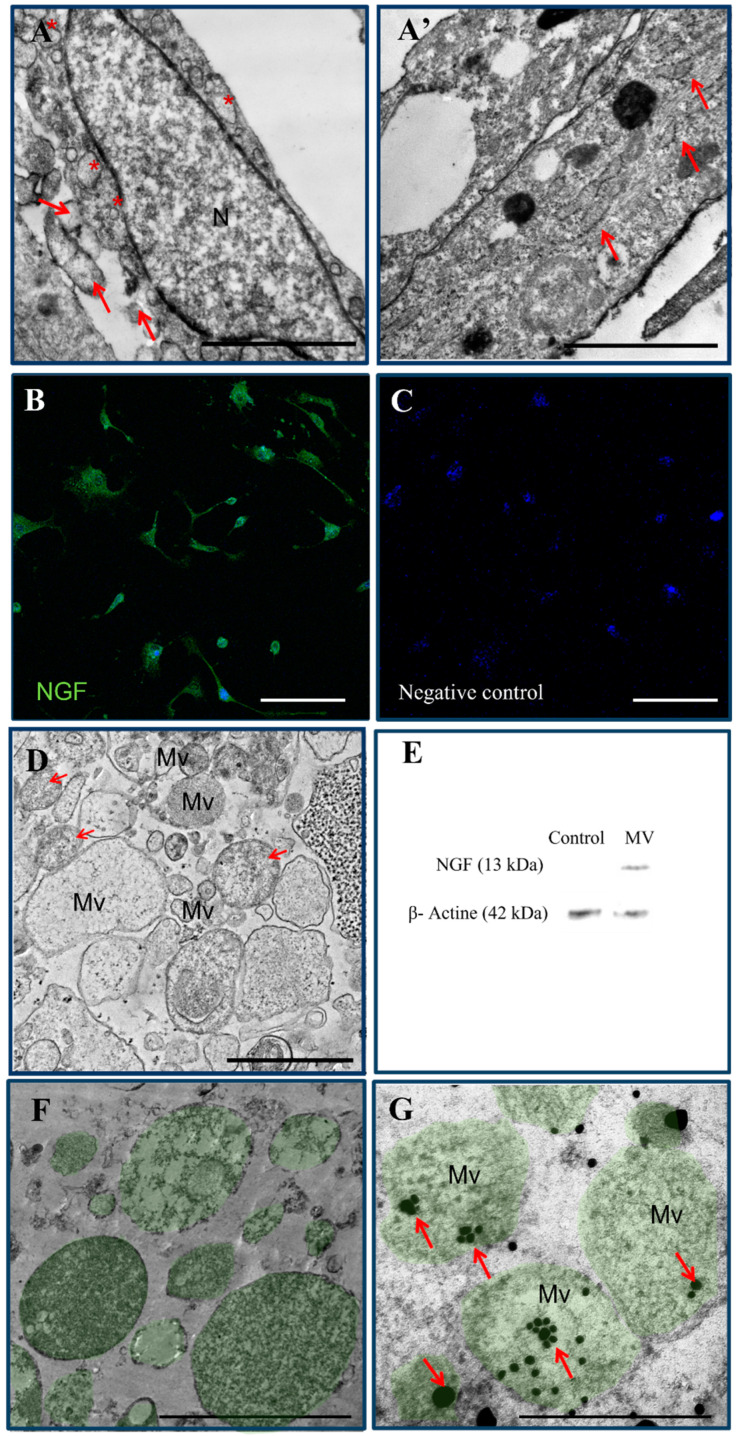
Characterization of ADMSC-MVs and analysis of NGF expression. (**A**) Ultrastructure of ADMSCs. The nucleus has an irregular elongated shape (N), there are many microvesicles in the cytoplasm (*), there are many natural exosomes in the surrounding space (arrow). (**A′**) ultrastructure of ADMSCs after transduction with NGF. (**B**,**C**) Confocal microscopy of ADMSCs after LV-NGF transduction. (**B**) NGF-positive ADMSCs. (**C**) Control of non-transduced ADMSCs. (**D**) Transmission electron microscopy of ADMSC microvesicles. Microvesicles are heterogeneous in size, have contents of different electron density, mitochondria are detected among them (arrow). (**E**) Western blot of ADMSC-MVs-NGF and ADMSC-MVs (Control). (**F**) NGF-negative reaction in the cytoplasm of microvesicles obtained from ADMSCs. (**G**) NGF-positive reaction in the cytoplasm of microvesicles obtained from LV-NGF-transduced ADMSCs (black dots are gold nanoparticles with a diameter of 5 nm, reinforced with silver attached to NGF primary antibody) (arrow). Scale: (**A**) 2 µm, (**B**) 200 µm, (**C**) 2 µm, (**D**,**F**,**G**) 1 µm.

**Figure 2 ijms-24-08332-f002:**
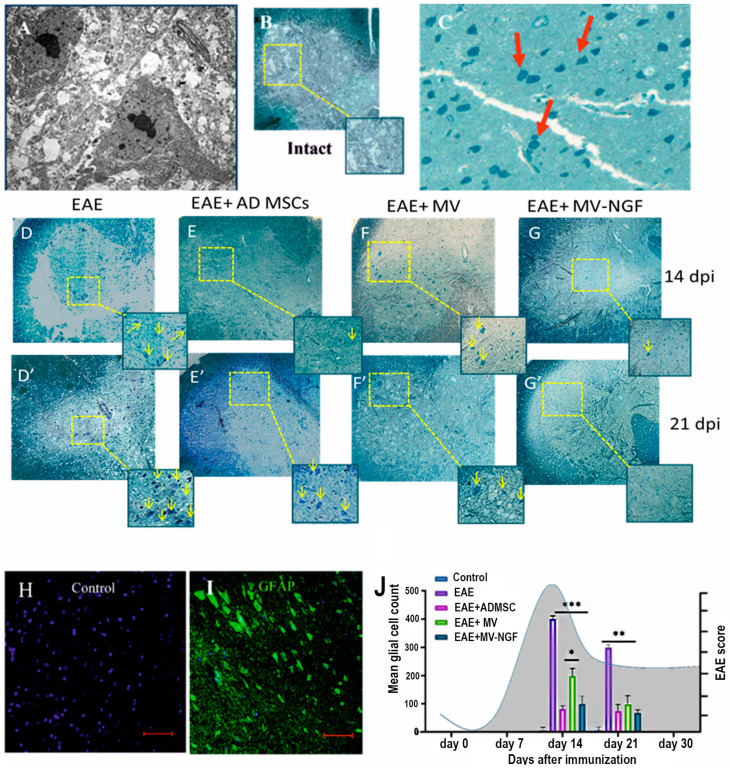
Immunohistochemistry was performed on the spinal cord tissues of mice with EAE without treatment to detect the phenotype of proliferated cells using GFAP as a structural marker of reactive astrocytes. These cells were positively stained for GFAP (**H**,**I**). Therefore, we hypothesize that these reactive glial cells are most likely reactive astrocytes. Morphological and quantitative manifestations of spinal cord neurogliosis in control and experimental groups. (**A**) Electron microscopy showing the ultrastructure of spinal cord tissue without treatment. (**B**) semi-thin section of intact mouse spinal cord, stained with toluidine blue, showing neurons with light cytoplasm (selected area), with no signs of neurogliosis. (**C**) Appearance of reactive glial cells in semi-thin sections stained with methylene blue in EAE (red arrows). Scale: 50 µm. (**D**–**G’**) Semi-thin sections of the spinal cord in EAE without treatment at 14 dpi (**D**–**G**) and 21 dpi (**D′**–**G′**). The sections showed large and irregularly shaped glial cells stained with toluidine blue (the highlighted area is shown with arrows pointing to glial cells). Scale segment. (**H**) Negative immunofluorescence reaction of glial cells to GFAP. Scale: 100 µm. (**I**) Positive immunofluorescence reaction of glial cells to GFAP. Scale: 100 µm. (**J**) Glial cell count in the gray matter of spinal cord tissues of experimental groups at 14 and 21 dpi. * *p* < 0.05,** *p* < 0.001,*** *p* < 0.0001.

**Figure 3 ijms-24-08332-f003:**
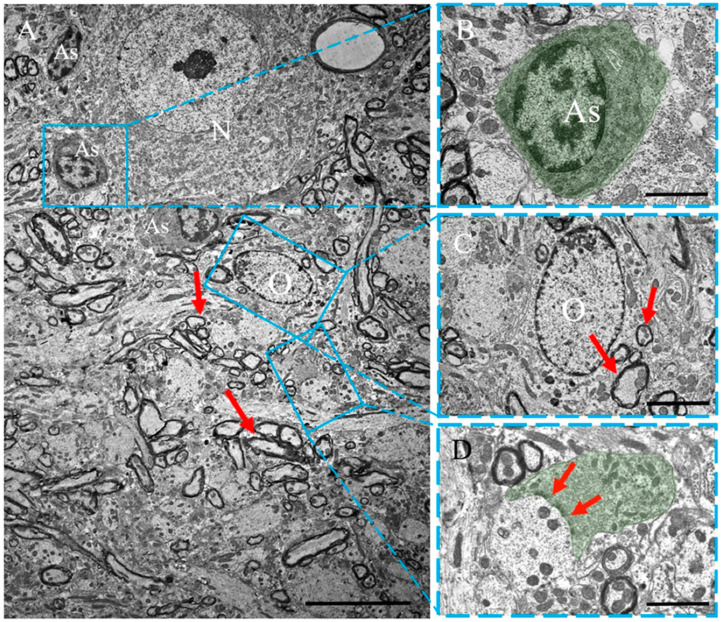
Ultrastructure of the gray matter of intact spinal cord (control animals). (**A**) General appearance: A motor neuron (N) with surrounding myelin (arrows) and myelin-free nerve fibers forming synaptic contacts, oligodendrocytes (O) and normal (inactive) astrocytes (As). (**B**) The area containing inactive astrocytes at high magnification (As): the cytoplasm area of the cell is highlighted green. (**C**) High magnification of the ultrastructure of an intact oligodendrocyte (O): myelin fibers are marked with an arrow. (**D**) High magnification of the synaptic contact (the presynaptic terminal is highlighted in green; the active areas are marked with an arrow). Scale: (**A**) 10 µm, (**B**–**D**) 2 µm.

**Figure 4 ijms-24-08332-f004:**
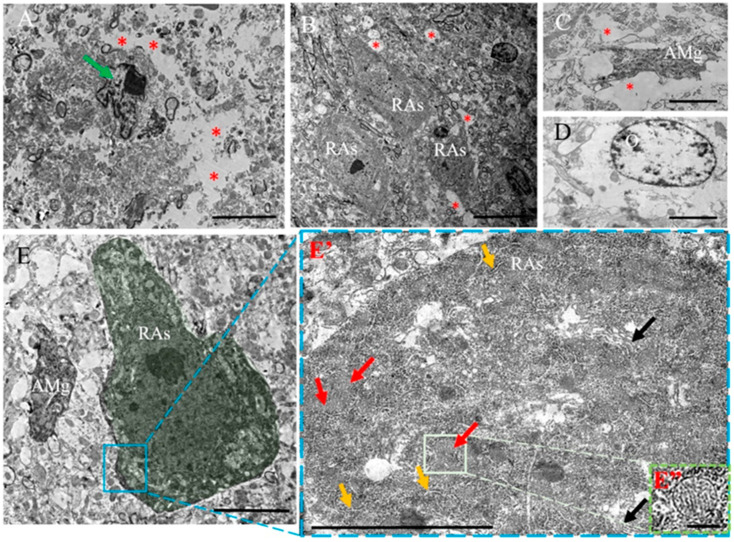
Ultrastructure of the gray matter of the EAE mouse spinal cord. (**A**) General appearance: large cavities (*) and degenerated cells with lobule nucleus (green arrow) on the 14th day after the formation of EAE. (**B**) Partially preserved area near a large cavity, general appearance: A group of reactive astrocytes (RAs) and partially degenerated nerve processes surrounding them (*) on the 21st day after the formation of EAE. (**C**) Reactive microglia cell (AMg) and partially degenerated nerve processes surrounding them (*) on the 14th day after the formation of EAE. (**D**) Degenerated oligodendrocyte (O) on the 21st day after the formation of EAE. (**E**) Ultrastructure of reactive astrocyte (cytoplasm highlighted in green) possesses pyramidal shape with electron-dense cytoplasm. (**E′**) The marked area of (**E**) at high magnification: well-developed and actively functioning organelles in the cytoplasm of a reactive astrocyte: Golgi complex (black arrow), rough ER (yellow arrow) and mitochondria (red arrow). (**E″**) Marked area: round mitochondria with developed cristae. Scale (**A**) 10 µm, (**B**–**D**) 20 µm, (**E**) 5 µm, (**E′**) 2 µm, (**E”**) 200 nm.

**Figure 5 ijms-24-08332-f005:**
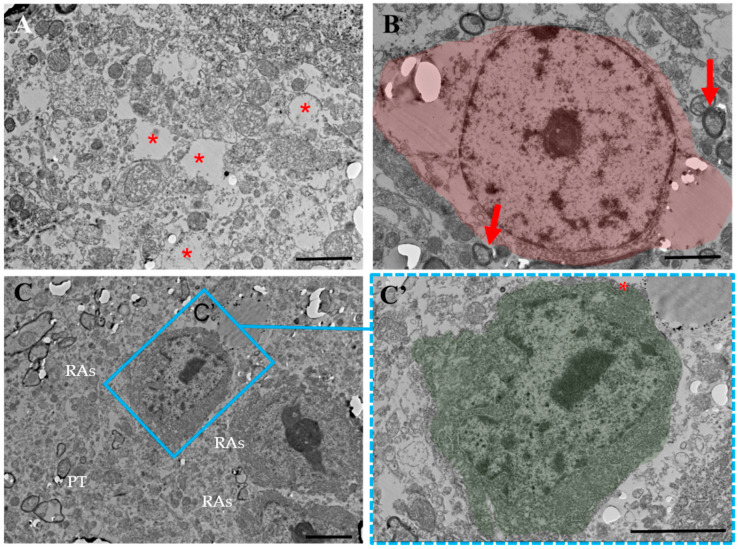
Ultrastructure of the gray matter of the spinal cord of an EAE mouse treated with ADMSCs. (**A**) Preserved nerve tissue with single small cavities from degenerated nerve fibers (*) at 14 dpi. Scale: 2 µm. (**B**) Preserved nerve tissue with areas of new remyelinating fibers (arrow) along with a preserved motor neuron (highlighted in pink) at 21 dpi. Scale: 2 µm. (**C**) Reactive astrocytes around a preserved tissue at 14 dpi create a scare. (**C′**) In selected regions, the astrocyte (highlighted in green) stayed in a border between preserved tissue and areas of degenerated nerve processes (*), Scale: 2 µm. Scale: 5 µm. RAs—reactive astrocytes, PT—preserved tissue.

**Figure 6 ijms-24-08332-f006:**
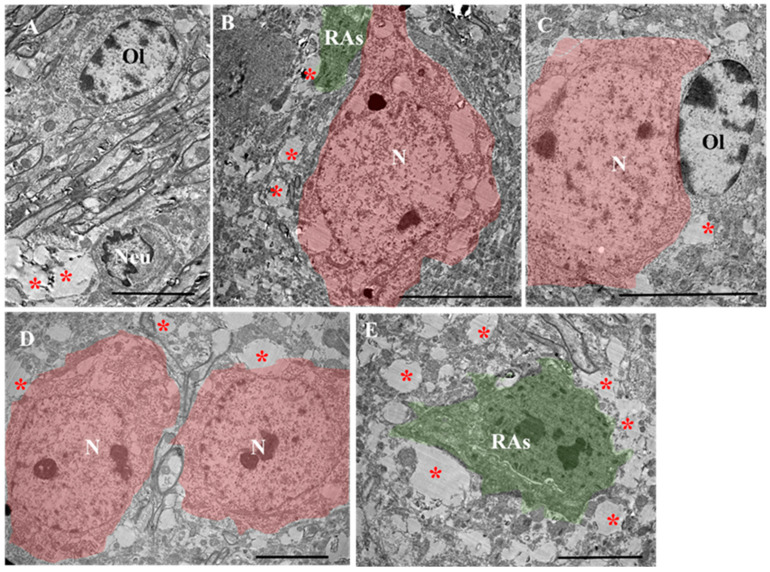
Ultrastructure of the gray matter of the spinal cord of an EAE mouse treated with ADMSC-MVs. (**A**) Preserved myelin fibers, oligodendrocytes (OL) and small, enlightened areas of degeneration in the area in which neutrophils (Neu) are present at 14 dpi. (**B**) Neuron (highlighted in red) in the initial stage of degeneration with vacuolized cytoplasm. The perikaryon is surrounded by cells with an electron-dense cytoplasm, probably reactive astrocytes (RAs, highlighted in green.). (**C**) A preserved neuron (highlighted in red) in a modified tissue, with an adjacent preserved oligodendrocyte (OL). The modified area shows a homogeneous preserved tissue along with small areas of degenerated nerve processes at 14 dpi. (**D**) Preserved neuron (highlighted in red) at 21 dpi. (**E**) Reactive astrocyte (RA, highlighted in green) in a modified tissue ((*) degenerated axons) at 21 dpi. Scale bar: in (**A**,**D**,**E**) 5 µm, (**B**,**C**) 10 µm.

**Figure 7 ijms-24-08332-f007:**
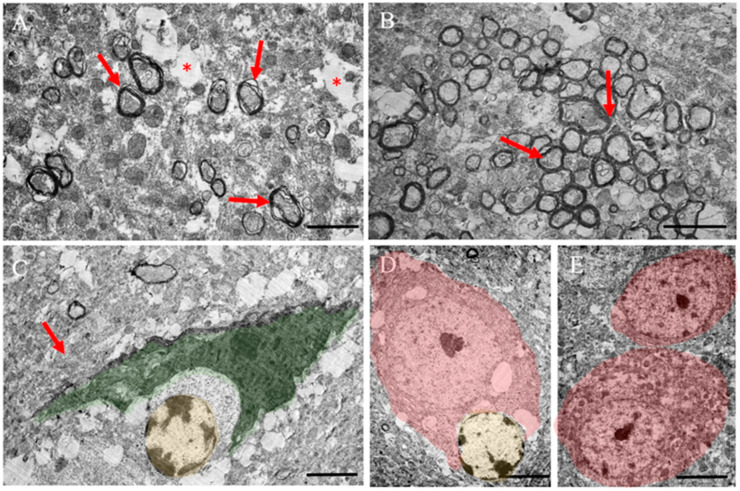
Ultrastructure of the gray matter of the spinal cord of an EAE mouse treated with ADMSC-MVs-NGF. (**A**) Preserved nerve tissue and separate newly formed myelin fibers (arrow), few single small cavities from degenerated nerve fibers (*) on day 14, after the formation of EAE. (**B**) Preserved nerve tissue and an area of new remyelinating fibers (arrow). (**C**) Reactive astrocyte (highlighted in green) in preserved-modified tissue (arrow). In the modified area, lymphocytes are identified (highlighted in yellow), homogeneous preserved tissue and small areas of degenerated nerve processes on the 14th day after the formation of EAE. (**D**) Preserved neuron (highlighted in pink) and lymphocyte (highlighted in yellow) on the 14th day after the formation of EAE. (**E**) Preserved motor neurons (highlighted in pink) on the 21st day after the formation of EAE. Scale segment is (**A**,**B**) 2 µm, (**C**) 10 µm, (**D**,**E**) 5 µm.

**Figure 8 ijms-24-08332-f008:**
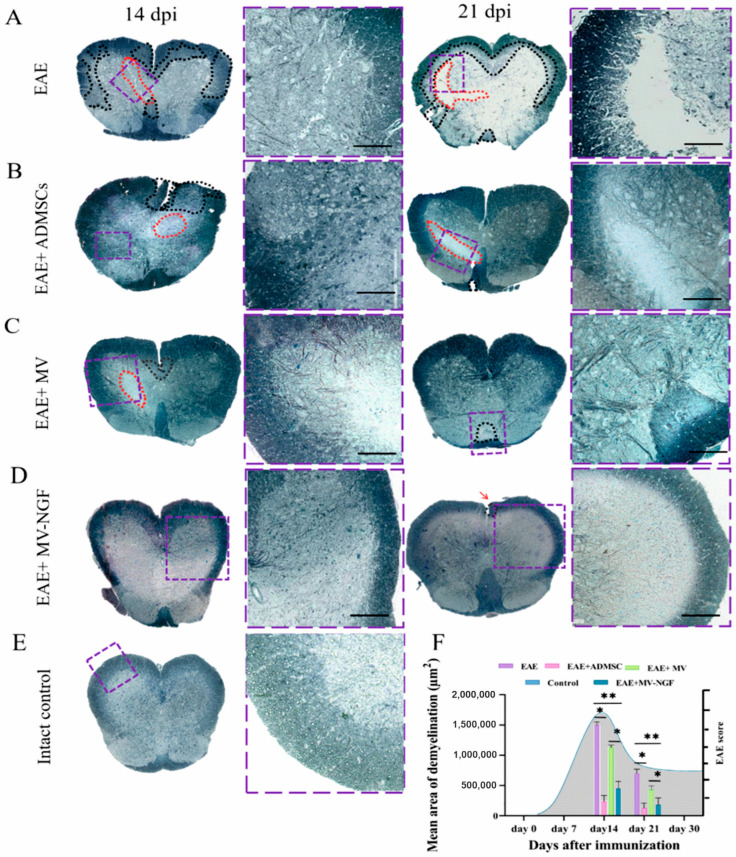
(**A**–**D**) Sections of the spinal cord of the studied groups of animals on the 14th and 21st days after the formation of EAE (stained with methylene blue). Areas of demyelination and destruction are surrounded with dashed black lines. Red dashed lining points to the cavities and damaged sites of the gray matter. Violet windows are extracted and displayed on high magnification (arrow). (**E**) Section of control intact spinal cord. (**F**) Calculations of the area of demyelination of the spinal cord tissues of the studied groups of animals on 14th and 21st days after the formation of EAE. * *p* < 0.05, ** *p* < 0.001. Area under the curve represents the EAE score. Scale: (**A**–**E**) 400 µm.

**Figure 9 ijms-24-08332-f009:**
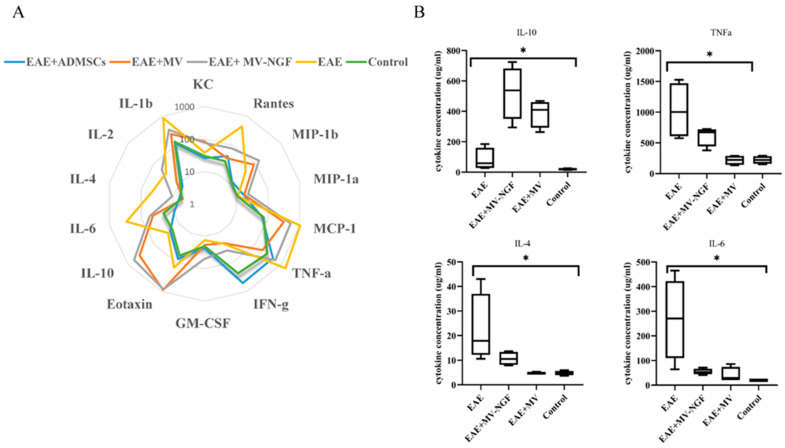
Cytokine levels in serum blood samples of experimental groups on the 21st day after the formation of EAE. (**A**) Diagram showing level of all investigated cytokines. (**B**) Diagram shows only the cytokines that play a critical role in the development of EAE. * *p* < 0.05.

**Figure 10 ijms-24-08332-f010:**
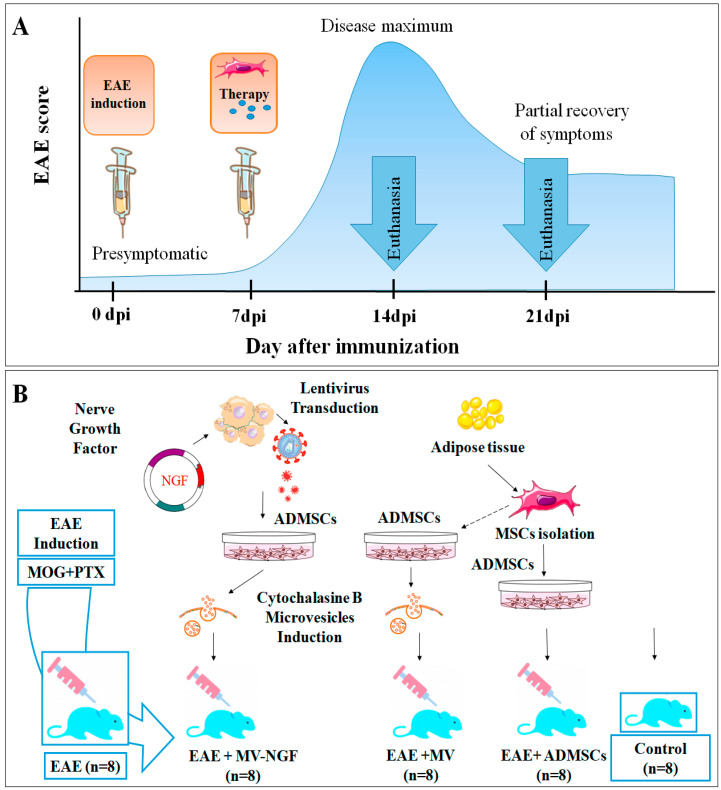
Study design. (**A**) Experimental timetable. The blue area under the curve represents the course of EAE. At 7 dpi, EAE starts to develop, at 14 dpi the disease severity reaches its peak and at 21 dpi EAE severity starts to decline. Accordingly, EAE was induced on day 0. Therapy was injected into animal groups on day 7 after EAE induction. Organs and blood serum samples were collected from mice (n = 8 in each experimental group) on 14th, 21st dpi. (**B**) Experimental animal groups. ADMSCs were isolated from mouse adipose tissue. ADMSCs were transfected with LV-NGF. Cytochalasine-B microvesicles were induced from ADMSCs. Animals were divided into 5 experimental groups (8 mice in each); each received a different type of treatment: negative control (EAE), ADMSCs, ADMSC-induced microvesicles (ADMSC-MVs), LV-NGF-ADMSC-induced microvesicles (ADMSC-MVs-NGF) and positive control. EAE—experimental autoimmune encephalomyelitis, ADMSCs—adipose derived mesenchymal stromal cells, MV—microvesicles.

**Table 1 ijms-24-08332-t001:** Primary and secondary antibodies used in Western blotting (WB) and immunofluorescence (IF) analysis.

Antibody/Reagent	Host	Dilution	Source
Glial fibrillary acidic protein (GFAP)	Mouse	1:200 (IF)	Santa Cruz
Nerve Growth Factor (NGF)	Rabbit	1:100 (WB)	Santa Cruz
Anti-mouse IgG conjugated to Alexa 488	Donkey	1:200 (IF)	Invitrogen (Waltham, MA, USA)
Anti-rabbit IgG conjugated to Alexa 647	Donkey	1:200 (IF)	Invitrogen
HRP-conjugated anti-rabbit IgG	Goat	1:1000 (WB)	Thermo Fisher Scientific

## Data Availability

All data generated or analyzed during this study are included in this published article. The data that support the findings of this study are available from the corresponding author upon request.

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
