# Peer review of "Genetically Engineered Artificial Microvesicles Carrying Nerve Growth Factor Restrains the Progression of Autoimmune Encephalomyelitis in an Experimental Mouse Model"

_ijms, 2023, doi:10.3390/ijms24098332_

Round 1
Reviewer 1 Report
The paper explores a potential new therapy for multiple sclerosis (MS). The therapy involves the use of adipose mesenchymal stem cells (ADMSCs) derived, cytochalasin B induced artificial microvesicles (MVs) expressing nerve growth factor (NGF) on a EAE model of MS. The study found that the injection of ADMSCs-MVs-NGF led to a significant decrease in neurogliosis post EAE induction and stimulated axon regeneration. Therefore, the paper concludes that artificial ADMSCs-MVs-NGF could be a new effective treatment for MS, modulating gliosis in the EAE model.
This manuscript is well-written, easy to follow, and clearly provides the authors’ view on this subject. I am confident that the manuscript will be well-received by the readers of IJMS.
Author Response
Esteemed Reviewer, we express our sincere gratitude for your careful review and consideration of our manuscript.
Reviewer 2 Report
The authors have developed ADMSCs derived, cytochalasin B induced artificial MVs , which expressing NGF (ADMSCs-MVs-NGF).
The characterization analysis showed that confirmation NGF expression of ADMSCs-MVs-NGF.
The therapeutic efficacy of ADMSCs-MVs-NGF was investigated in a mouse model of multiple sclerosis experimental auto-immune encephalomyelitis (EAE).
The injection of ADMSCs-MVs-NGF into EAE mice showed a significant decrease in neurogliosis.
Comments
1) The authors may include the scale of symptoms of EAE as a graph.
2) Was cytokine profile evaluated on 21 p.i.? Please indicate the day of serum was collected.

Author Response
Dear Reviewer:
Thank you very much for having considered our manuscript. We are very happy to have received a positive evaluation, and we would like to express our appreciation to you. We added recommended changes in the current paper.